# A ferritin-based COVID-19 nanoparticle vaccine that elicits robust, durable, broad-spectrum neutralizing antisera in non-human primates

Payton A.-B. Weidenbacher [1,2,12], Mrinmoy Sanyal [1,3,12], Natalia Friedland[1,3,12], Shaogeng Tang[1,3], Prabhu S. Arunachalam [4], Mengyun Hu [4], Ozan S. Kumru [5], Mary Kate Morris[6], Jane Fontenot[7], Lisa Shirreff [7], Jonathan Do[1,3], Ya-Chen Cheng[1,3], Gayathri Vasudevan[8], Mark B. Feinberg[8], Francois J. Villinger[7], Carl Hanson[6], Sangeeta B. Joshi[5], David B. Volkin[5], Bali Pulendran [4,9,10] & Peter S. Kim [1,3,11] ✉

While the rapid development of COVID-19 vaccines has been a scientific triumph, the need remains for a globally available vaccine that provides longer-lasting immunity against present and future SARS-CoV-2 variants of concern (VOCs). Here, we describe DCFHP, a ferritin-based, protein-nanoparticle vaccine candidate that, when formulated with aluminum hydroxide as the sole adjuvant (DCFHP-alum), elicits potent and durable neutralizing antisera in non-human primates against known VOCs, including Omicron BQ.1, as well as against SARS-CoV-1. Following a booster ~one year after the initial immunization, DCFHP-alum elicits a robust anamnestic response. To enable global accessibility, we generated a cell line that can enable production of thousands of vaccine doses per liter of cell culture and show that DCFHP-alum maintains potency for at least 14 days at temperatures exceeding standard room temperature. DCFHP-alum has potential as a once-yearly (or less frequent) booster vaccine, and as a primary vaccine for pediatric use including in infants.

The COVID-19 pandemic was met with record-breaking vaccine development speed[1,2], and widespread vaccination is estimated to have prevented over 14 million deaths in the first year of implementation[3]. Nonetheless, there remains an urgent public health need for vaccine interventions. First, as of May, 2022, the WHO estimates that almost one billion people globally remain unvaccinated against SARS-CoV-2[4].

Second, affordability continues to be an obstacle to global vaccine accessibility, including the cost of low-temperature storage and transport[5,6]. Third, the protection against infection provided by vaccine-induced or infection-induced immunity wanes with time, which has led to frequent booster vaccine doses[7]. Fourth, SARS-CoV-2 variants of concern (VOCs) capable of evading natural or vaccine-

[1]Sarafan ChEM-H, Stanford University, Stanford, CA, USA. [2]Department of Chemistry, Stanford University, Stanford, CA, USA. [3]Department of Biochemistry, School of Medicine, Stanford University, Stanford, CA, USA. [4]Institute for Immunity, Transplantation and Infection, Stanford University School of Medicine, Stanford, CA, USA. [5]Vaccine Analytics and Formulation Center, Department of Pharmaceutical Chemistry, University of Kansas, Lawrence, KS, USA. [6]California Department of Public Health, Richmond, CA, USA. [7]New Iberia Research Center, University of Louisiana at Lafayette, New Iberia, LA, USA. [8]IAVI, New York, NY, USA. [9]Department of Pathology, Stanford University School of Medicine, Stanford, CA, USA. [10]Department of Microbiology and Immunology, Stanford University School of Medicine, Stanford, CA, USA. [11]Chan Zuckerberg Biohub, San Francisco, CA 94158, USA. [12]These authors contributed equally: Payton A.-B. Weidenbacher, Mrinmoy Sanyal, Natalia Friedland. ✉e-mail: kimpeter@stanford.edu

induced humoral immunity continue to emerge[8]. Finally, as SARS-CoV-2 becomes endemic, worldwide immunization of the pediatric population, including of infants, remains an unmet need[9].

Protein nanoparticle vaccines, compared to isolated protein subunits, are more readily taken up by antigen-presenting dendritic cells[10,11] and the multivalent presentation of the antigen facilitated by the nanoparticles promotes receptor clustering and subsequent activation of B cells[12,13]. Indeed, candidate ferritin-based nanoparticle vaccines have shown robust humoral immune responses against SARS-CoV-2[14–16] and other viral glycoproteins[17–21], including showing safety and efficacy in clinical trials[20,22]. Previously, we introduced a protein nanoparticle-based vaccine candidate, SΔC-Fer, which displays a truncated form of the prefusion SARS-CoV-2 spike-protein ectodomain trimer from the Wuhan-1 isolate on self-assembling *Heliobacter pylori* ferritin nanoparticles[14]. SΔC-Fer contains an inactivated polybasic cleavage site, which has been shown to improve neutralizing titers[23], and the 2-proline (2P)[24] prefusion-stabilizing substitutions found in the FDA-approved SARS-CoV-2 mRNA vaccines[1,25]. Importantly, SΔC-Fer also contains a deletion of 70 amino acid residues from the C-terminus of the spike ectodomain. This deletion removes a highly flexible region that is not well resolved in cryo-EM structures[26–28], and that contains immunodominant, linear (i.e., not conformational) epitopes frequently targeted by antibodies in convalescent COVID-19 plasma[29,30]. Removal of these immunodominant linear epitopes and multivalent presentation of the modified spike protein on a ferritin nanoparticle substantially improved the neutralizing potency of elicited antisera relative to other tested vaccines in mice[14].

Here, we introduce an updated version of SΔC-Fer, called Delta-C70-Ferritin-HexaPro or DCFHP. We supplemented the 2P stabilizing substitutions with four previously described[27] proline substitutions to create a six-proline substituted (HexaPro) version of the vaccine. Previous work has shown that the HexaPro SARS-CoV-2 spike protein has increased stability and expression relative to the 2P version[27] further differentiating this vaccine from previous spike-ferritin products[16,31]. We found that DCFHP is more stable to thermal denaturation than SΔC-Fer. In addition, we show that DCFHP can be generated in a Chinese hamster ovary (CHO) cell line at levels exceeding 2 grams per liter.

Our vaccine formulation, DCFHP-alum, consists of DCFHP antigen formulated with aluminum hydroxide (Alhydrogel™, referred to herein as alum) as the only adjuvant[32]. Aluminum salt adjuvants are the most commonly used adjuvant in human vaccines licensed by the FDA and regulatory agencies worldwide, and have been administered to billions of individuals over the past 90 years[33–35]. Moreover, aluminum salt adjuvants are currently used in infant vaccines against hepatitis B, diphtheria-tetanus-pertussis (DTaP), Haemophilus influenzae type b (Hib), and pneumococcus infectious agents[36], with an excellent safety profile[37,38].

We show here that DCFHP-alum elicits a robust and durable immune response in mice against SARS-CoV-2 VOCs. In addition, we demonstrate that DCFHP-alum remains stable at temperatures ranging from 4 °C to 37 °C for at least 14 days, as assessed by immunization studies in mice. Thus, we anticipate that local distribution of the DCFHP-alum vaccine could be feasible without refrigeration.

Finally, we demonstrate that a two-dose intramuscular immunization regimen in rhesus macaques with DCFHP-alum elicits antisera with durable, robust, and broad neutralization of VOCs, including the Omicron subvariants BA.4/5[39] and BQ.1[40], along with a balanced Th1 and Th2 immune response. Strikingly, these non-human primate (NHP) antisera also show robust and durable neutralization activity against the phylogenetically divergent SARS-CoV-1 pseudovirus[41]. Boosting these immunized NHPs after ~1 year with a third dose of DCFHP-alum elicits a robust, broad-spectrum, anamnestic neutralizing antibody response. Taken together, these results suggest that DCFHP-alum may provide an affordable and effective solution to pediatric and worldwide vaccination against SARS-CoV-2 and present the possibility of an effective, once-yearly (or less frequent) booster.

## Results

### Vaccine design and characterization

We sought to further optimize our ferritin-based nanoparticle vaccine, SΔC-Fer[14], to generate DCFHP, which includes additional stabilizing proline residues to promote robust expression. DCFHP maintains the 2P substitutions[24] and deletion of 70 C-terminal spike ectodomain residues contained in SΔC-Fer[14], but also incorporates four proline residue substitutions and a modification to the mutated furin cleavage site, as described in the HexaPro spike design[27] (Fig. 1A). As expected, DCFHP is expressed at higher levels than SΔC-Fer following transient transfection in Expi293F cells (SI Fig. 1A, B). SDS-PAGE analysis of purified DCFHP showed the expected monomeric molecular weight of ~160 kDa (Fig. 1B). In addition, size-exclusion chromatography coupled with multiangle light scattering (SEC-MALS) analysis shows that the nanoparticle has a molecular weight (~3.4 MDa) consistent with a ferritin-based nanoparticle displaying eight copies of the SARS-CoV-2 spike ectodomain trimer (Fig. 1C). Differential scanning fluorimetry (DSF) experiments indicate that thermal denaturation of DCFHP occurs at higher temperatures compared to SΔC-Fer, and these changes are similar to those observed between the 2P and HexaPro variants of the spike trimer[27] (SI Fig. 1C), suggesting that HexaPro mutations induce similar stabilizing effects in the context of the nanoparticle. Biolayer interferometry (BLI) binding of conformation-specific antibodies[42–45] indicates proper epitope presentation (SI Fig. 1D). Finally, single-particle cryo-electron microscopy (cryo-EM) of DCFHP shows a multivalent particle displaying eight copies of the SARS-CoV-2 trimer arrayed radially from a ferritin core (Fig. 1D, SI Fig. E, F), as was observed previously with SΔC-Fer[14]. Collectively, these data suggest that the spike component of DCFHP maintains the same native conformation as in SΔC-Fer, with increased stability and expression levels conferred by the HexaPro substitutions.

Finally, mice immunized with either SΔC-Fer or DCFHP, adjuvanted with a high dose of alum and CpG, showed highly robust and similar immunogenic profiles by enzyme-linked immunosorbent assay (ELISA) (SI Fig. 2A) and pseudoviral neutralization assays[46] (Fig. 1E); see also ref. 32.

### DCFHP formulated with alum is highly immunogenic and stable at elevated temperatures

Given the robust immunogenicity of DCFHP formulated with alum and CpG, we hypothesized that the use of alum as the sole adjuvant for DCFHP (i.e., DCFHP-alum) might still elicit a robust response. Indeed, though the response was diminished compared to alum + CpG, DCFHP-alum elicited strong pseudoviral neutralizing titers of approximately $10^4$ in mice (SI Fig. 2B). Under these conditions, DCFHP is 100% bound to alum[32].

To investigate the stability of the DCFHP-alum vaccine, samples were stored at 4 °C, 27 °C, or 37 °C for varying times and the immunogenicity of these stored samples was evaluated in a single-dose mouse immunization study. Remarkably, the DCFHP-alum vaccine remained similarly immunogenic across all temperatures and storage periods, as measured in pseudoviral neutralization assays (Fig. 2). We conclude that DCFHP-alum is stable, as measured by mice immunogenicity studies, after storage for at least two weeks at 37 °C.

### Stable expression of DCFHP increases yield and simplifies purification

To facilitate future manufacturing of DCFHP under good manufacturing practices (GMP) conditions, we next aimed to develop a high-producing stable mammalian cell line expressing DCFHP. To do so, multiple copies of a codon-optimized DCFHP gene were inserted into CHO-K1 cells using a Leap-In transposase at ATUM[47,48], a

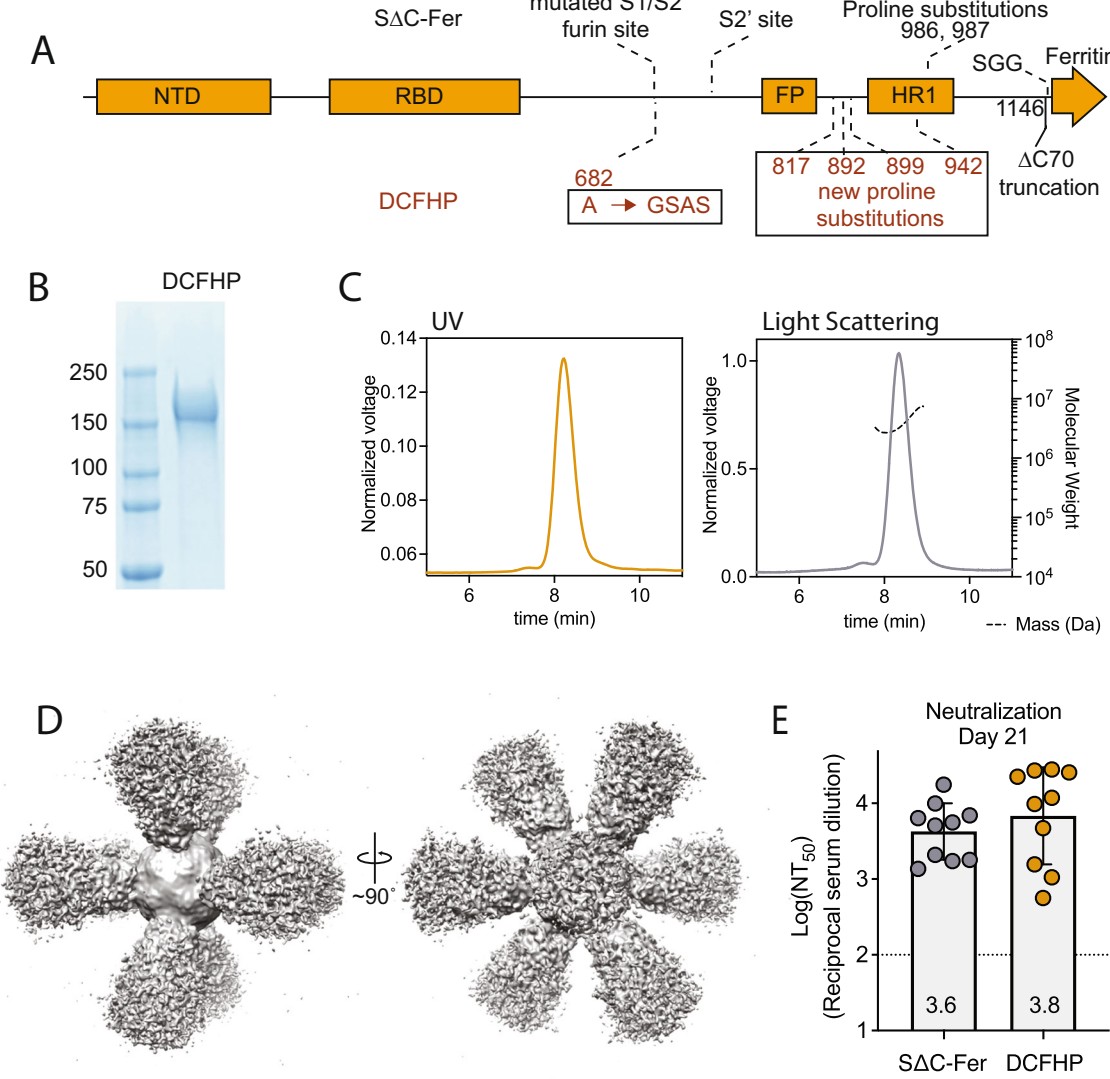

**Fig. 1 | DCFHP design and validation. A** DCFHP schematic showing the modifications made to convert S∆C-Fer into DCFHP in red. Receptor binding domain (RBD), N-terminal domain (NTD), S1/S2 cleavage site, S2' cleavage site, fusion peptide (FP), heptad repeat 1 (HR1), as annotated. **B** SDS-PAGE gel showing purified DCFHP running as a monomer at the anticipated kDa molecular weight (ladder, shown left). The gel is representative of the numerous DCFHP samples run by SDS-PAGE. **C** UV (yellow) and light scattering (gray) traces determined from SEC-MALS shows a homogenous nanoparticle peak with approximate molecular weight (dashed line) of 3.4 MDa. **D** 3D reconstructed cryo-EM density maps of DCFHP, refined with octahedral symmetry. **E** Similar robust neutralization of Wuhan-1 SARS-CoV-2 pseudovirus with day 21 serum from mice immunized with either S∆C-Fer or DCFHP formulated with 500 µg alum and 20 µg CpG 1826, following a single immunization. Neutralization titers were assessed in a HeLa cell line expressing ACE2 and TMPRSS2. Data for 10 mice are presented as geometric mean titer and standard deviation. Assay limit of quantitation (LOQ) is shown as a dotted horizontal line.

bioengineering company (see Methods). As anticipated[49], N-glycosylation profiles of these proteins expressed transiently in Expi293 versus stably in CHO cells differ[32] but they are similarly immunogenic in mice, as measured in pseudoviral neutralization assays[32]. Following preliminary analysis of expression in pooled cells, 125 single-cell clones were evaluated for high levels of cell growth and productivity, resulting in 24 lead clones (SI Fig. 3A–C). Among these, we chose clones (SI Fig. 4A–C) that showed favorable expression and nanoparticle assembly, as determined by SDS-PAGE, biolayer interferometry (BLI), and SEC-MALS (SI Fig. 3A–C, respectively). In this manner, we identified five CHO-K1 stable cell clones (SI Fig. 4A) that express DCFHP at a high nanoparticle:monomer ratio (SI Fig. 4B), with expected stability (SI Fig. 4C), and exceptional yield (>2 g/L, SI Fig. 4A).

In addition, we were able to optimize the purification of the nanoparticles over our previous method[14]. The CHO cell-derived supernatant was supplemented to 200 mM sodium chloride and flowed over a HiTrapQ anion-exchange column. The nanoparticle-containing flowthrough was concentrated and subjected to size exclusion chromatography purification (see Methods). These changes enabled rapid and simplified nanoparticle purification, with increased yield, while maintaining high overall purity.

## DCFHP-alum is robustly immunogenic in non-human primates (NHPs)

Encouraged by results demonstrating that sera from mice immunized with DCFHP-alum showed potent neutralizing activity against SARS-CoV-2 pseudovirus, we sought to investigate the immunogenicity of the DCFHP-alum vaccine in NHPs. The NHPs are an antigen-naïve population, acting as a surrogate for a pediatric population. DCFHP purified from engineered CHO-K1 cells was used for the immunization of ten male rhesus macaques, aged between 3 and 9 years (SI Table 1). The ten NHPs were divided into two groups to test the effect of a short and a long gap between the primary and booster doses. Longer gaps

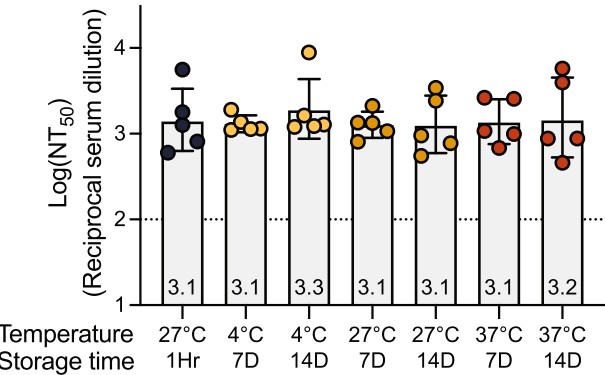

**Fig. 2 | Formulated DCFHP-alum is thermostable up to 37 °C for 14 days.** Neutralization titers against Wuhan-1 SARS-CoV-2 pseudovirus for serum obtained from individual animals 42 days following immunization with DCFHP-alum (10 μg DCFHP with 150 μg alum) that had been stored at a range of temperatures (bottom) for either 7 days or 14 days, compared to freshly formulated DCFHP-alum (left, black circles). Assay limits of quantitation is shown as a dotted horizontal line. A single representative experiment of samples run in technical duplicate is shown. GMT and STD are shown.

have been found to produce more robust immune responses in other vaccines[50].

We primed both NHP groups on day 0 and boosted at either day 21 (group A) or day 92 (group B) (Fig. 3A and SI Table 2). DCFHP-alum elicited a neutralizing immune response 21 days after a single immunization in both groups (Fig. 3B). While the response was substantially improved following a booster dose in both groups, the delayed boost elicited better neutralizing antisera against variants, as assessed 14 days post boost (Fig. 3C, D; SI Table 3). Interestingly, while group A elicited a more robust response against Wuhan-1 (SI Fig. 5), on average, group B showed approximately a 4-fold increased neutralizing response relative to group A against divergent VOCs (SI Fig. 5). Remarkably, DCFHP-alum vaccination of NHPs also elicits robust neutralization of pseudotyped SARS-CoV-1 (Fig. 3C, D), which is distant from SARS-CoV-2 on the sarbecovirus (SARS-like betacoronavirus) phylogenetic tree[41].

When tested against authentic SARS-CoV-2 virus, consistent with the results in pseudoviral assays, sera from the NHPs showed neutralization against the Wuhan-1 virus and VOCs (Fig. 3E). Notably, a ferritin-based SARS-CoV-2 vaccine candidate that elicits similar neutralizing titers to those described here has been shown to confer protection in hamster[51] and NHP challenge models[16].

**Long-lived immune responses in non-human primates**
We further investigated the durability of the vaccine-induced neutralizing responses elicited in groups A and B (Fig. 4A–D, SI Fig. 6, SI Table 2). Notably, all NHPs maintained a neutralizing antiserum response against the Wuhan-1 pseudovirus persisting for at least 250 days (Fig. 4A, C). Similarly, albeit more modestly, most animals in group B retained detectable neutralizing potency against BA.4/5 and the sequence-divergent SARS-CoV-1 out to ~one year (Fig. 4D and SI Fig. 6B), with titers generally higher than seen in group A (Fig. 4B, D and SI Fig. 6A, B).

Simple modeling suggests a biphasic decay of neutralizing activity against Wuhan-1 pseudovirus following the boost for group A (days 35–337) with an initial, fast-phase (half-life of ~5 weeks) followed by a slow-phase (half-life of several years) (Fig. 4, SI Fig. 7B and SI Table 4). The fast-phase accounts for approximately 50% of the total decay (SI Fig. 7B and SI Table 4). Simple modeling of these data with a single-phase decay shows a half-life of ~one year, with a poorer fit than for biphasic decay (SI Fig. 7A and SI Table 4). Similar trends are seen following the boost for group B (days 106–337), although with greater

variability, presumably because our long-term data for group B is sparse (SI Table 4). In either case, a substantial portion of the serum neutralizing activity appears to decay very slowly with time following immunization of NHPs with DCFHP-alum.

To explicitly test the potential of DCFHP-alum as an annual vaccine, we gave a second boost to all NHPs on day 381. Animals in both groups A and B showed a strong anamnestic immune response with $NT_{50}$ values against Wuhan-1, BA.4/5, SARS-CoV-1, and BQ.1 of approximately $10^4$, $10^{3.5}$, $10^3$, and $10^3$, respectively (Fig. 5A–H, data for BQ.1.1 shown in SI Fig. 8). The responses for the animals in group A is particularly striking given that their responses had generally waned prior to the second boost (compare Fig. 4A, B and SI Fig. 6A to Fig. 5A–C). Boosting after one year essentially abrogated the differences seen between groups A and B.

As seen previously for immune responses with protein-based vaccines in a naive population[52], we observed a dominant CD4 + T cell response with no detectable CD8 + T cell response. In both groups, DCFHP-alum elicited a balanced distribution of Th1 and Th2 CD4 + T cells (Fig. 6A–D, and SI Fig. 9). This response was similar to those reported for previous COVID-19 nanoparticle vaccines and with other adjuvant formulations[52] and mitigates the Th2 skewing seen in previous Alum-based vaccines[53]. Importantly, we detected responses from peptides derived not only from the original Wuhan-1 strain (Fig. 6A, C), but also from the Omicron BA.1 strain (Fig. 6B, D), suggesting responses targeting conserved T cell epitopes.

## Discussion
Our DCFHP-alum vaccine candidate, despite being based solely on the ancestral Wuhan-1 sequence, can elicit a robust, broad-spectrum neutralizing antiserum response in NHPs against SARS-CoV-2 VOCs, as well as against SARS-CoV-1, that is durable for >250 days. These findings challenge the notion that bivalent vaccines are required to address the COVID-19 pandemic. In addition, while there is continued discussion surrounding annual COVID-19 vaccine boosters[54-56], little to no work has been done to test the capacity of vaccines to elicit neutralizing antibody responses with durability of ~one year and to elicit a subsequent response following an annual booster dose. Our results show that DCFHP-alum can elicit durable, broad-spectrum neutralizing antisera (including against BA.4/5, BQ.1, and SARS-CoV-1) in NHPs. We also show that a robust, anamnestic response can be elicited following a second boost after ~one year.

Several studies of COVID-19 vaccines have established that a cell-culture pseudovirus neutralizing titer of ~$10^2$ translates into human vaccine efficacy against symptomatic disease of ~90%[55-59]. In addition, clinical trials have established a correlation between anti-SARS-CoV-2 monoclonal antibody levels (i.e., humoral immunity alone) and protection from COVID-19[60,61]. Indeed, SARS-CoV-2 variant booster vaccines have been accepted by the FDA and EMA for emergency use authorization using neutralizing antibody titer as a correlate of protection[62,63].

Accordingly, if our NHP results are reproduced in clinical trials, we anticipate that DCFHP-alum could be widely used as a booster vaccine for individuals who have been previously vaccinated with other COVID-19 vaccines, and in unvaccinated individuals that have been previously infected with SARS-CoV-2. Together, these two categories encompass a large fraction of the world's population. As such, DCFHP-alum would be an attractive annual (or less frequent) vaccine that could circumvent the need for continual variant chasing.

We also envision that DCFHP-alum could enable global access to COVID-19 vaccination given its low projected cost of goods, high scale of production, and favorable broad-spectrum profile. Our results show that a DCFHP stably integrated CHO-based cell line could enable low-cost, large-scale production. Assuming a vaccine dose of ≤100 micrograms and a purification yield of ≥10%, we anticipate production of thousands of DCFHP vaccine doses per liter of engineered CHO-K1 cell

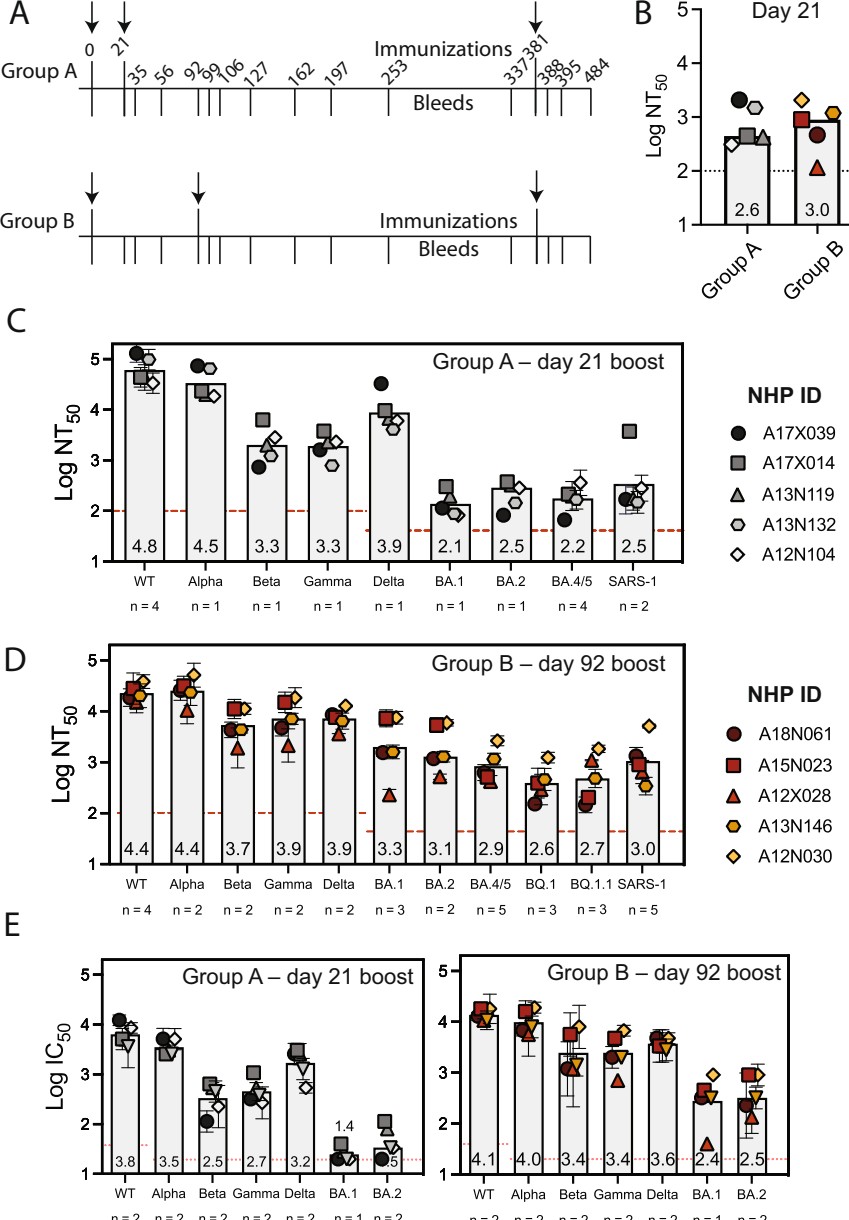

**Fig. 3 | DCFHP-alum immunized NHPs elicit cross-reactive neutralizing responses. A** Immunization scheme for NHPs immunized in either group A or group B with a 50 μg dose of DCFHP formulated with 750 μg alum (SI Table 2). Arrows indicate days of immunization. **B** Pseudoviral neutralization (plotted as the log of the neutralizing titer (reciprocal serum dilution)) of Wuhan-1 SARS-CoV-2 from NHP serum obtained 21 days following initial immunization are similar between groups A and B. **C** Cross-reactive pseudoviral neutralization by serum from NHPs isolated 14 days post boost. Means and standard deviations for biological replicates are plotted and noted for each animal (*n* = number of replicate neutralization assays conducted for these samples on independent days). **D** As in panel (**C**) for group B. **E** Limited dilution, neutralization values for authentic SARS-CoV-2 VOCs for serum samples obtained 14 days post-boost. NHP identification provided correlate with SI Table 1. Assay limits of quantitation indicated by horizontal dotted line. *n* is defined as individual replicates of the experiment, GMT and STD are shown.

culture. As well, the DCFHP-alum formulation is stable for at least two weeks at temperatures exceeding standard room temperature. Taken together, DCFHP-alum is an excellent candidate for development as a new COVID-19 vaccine, offering broad-spectrum protection, and providing the opportunity for global access without the cold chain distribution challenges encountered with other vaccines[64].

Finally, we anticipate the potential use of DCFHP-alum as an important primary vaccine in previously unvaccinated and uninfected individuals, especially in pediatric populations, including infants. Aluminum salt adjuvants are commonly used in infant vaccines and as part of routine childhood immunization schedules, and their excellent safety profile has been established over decades[33-38]. In infants and

other DCFHP-alum vaccine recipients naive to SARS-CoV-2 infection or vaccination, we would anticipate robust, cross-reactive responses similar to the naive NHPs studied here. Since primary immunization of NHPs with DCFHP-alum provides remarkably broad protection against VOCs, DCFHP-alum may be an ideal way to establish initial immune imprinting[65-67] against SARS-CoV-2 in infants.

## Methods
### IgG plasmids
Antibody sequences and Fc-tagged ACE2 were cloned into the CMV/R plasmid backbone for expression under a CMV promoter. The antibodies with variable HC/LC were cloned between the CMV promoter

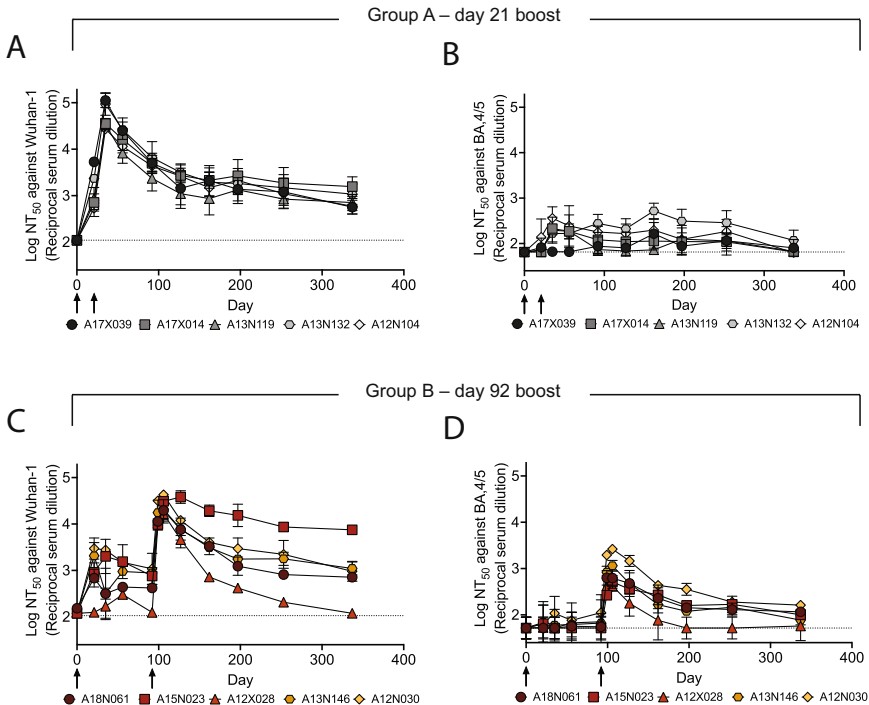

**Fig. 4 | DCFHP-alum immunized NHPs elicit long-lived immunity against both Wuhan-1 and BA.4/5 pseudoviruses. A** Serum neutralizing titers were monitored over 337 days against Wuhan-1 SARS-CoV-2 pseudovirus for animals in group A; days of prime and boost indicated with arrows. **B** As in panel (**A**) but against BA.4/5 pseudovirus. **C**, **D** As in panels (**A**) and (**B**) but with group B NHPs, respectively.

Averages and standard deviations for replicate neutralization assays are shown; for panels **A**–**D**: $n = 3$, $n = 4$, $n = 2$, and $n = 3$, respectively ($n$ = number of replicate neutralization assays conducted for these samples on independent days). NHP identification provided correlate with SI Table 1. Assay limits of quantitation indicated by horizontal dotted lines.

and the bGH poly(A) signal sequence of the CMV/R plasmid to facilitate improved protein expression. The variable region was cloned into the human IgG1 backbone. This vector also contained the HVM06_Mouse (P01750) Ig HC V region 102 signal peptide to allow for protein secretion and purification from the supernatant.

### Lentivirus plasmids

The 21-amino acid C-terminally truncated spike proteins with native signal peptides were cloned in place of the HDM-SARS2-spike-delta21 gene (Addgene plasmid, 155130). This construct contains a 21-amino acid C-terminal deletion to promote viral production, contained in all SARS-CoV-2 variants of concern. The SARS-CoV-1 spike contained an 18-amino acid C-terminal deletion. The other viral plasmids that were used have been previously described[46], including pHAGE-Luc2-IRES-ZsGreen (NR-52516), HDM-Hgpm2 (NR-52517), pRC-CMV-Rev1b (NR-52519) and HDM-tat1b (NR-52518).

### Other plasmids

An in-house pADD2 vector was used for all nanoparticle production. Sequences encoding DCFHP (residues 1–1146 of HexaPro)[27] and SΔC-Fer (residues 1–1143 as previously described)[14] were cloned into the pADD2 vector backbone using HiFi PCR (Takara) followed by In-Fusion (Takara) cloning with EcoRI/XhoI cut sites. This was followed by an amplicon containing H. pylori ferritin (residues 5–168) originally generated as a gene-block fragment from Integrated DNA Technologies (IDT). The spike and ferritin subunits were separated by a SGG linker as previously described[68].

The SARS-CoV-2 RBD construct was kindly provided by Dr. Florian Krammer[69]. It contains the native signal peptide (residues 1–14) followed by residues 319–541 from the SARS-CoV-2 Wuhan-Hu-1 genome sequence (GenBank MN908947.3). There is a C-terminal hexa-histidine tag. The pCAGGS plasmid contains a CMV promoter for protein expression in mammalian cells.

### Protein production

Transiently expressed proteins were expressed in Expi293F cells. Expi293F cells were cultured in media containing 66% Freestyle/33% Expi media (Thermo Fisher Scientific) and grown in TriForest polycarbonate shaking flasks at 37 °C in 8% CO$_2$. The day before transfection, cells were harvested by centrifugation and resuspended to a density of $3 \times 10^6$ cells/mL in fresh media. The next day, cells were diluted and transfected at a density of approximately $3–4 \times 10^6$ cells/mL. Transfection mixtures were made by combining the following components and adding the mixture to cells: maxi-prepped DNA, culture media, and FectoPro (Polyplus) at a ratio of 0.5–0.8 µg:100 µl:1.3 µl to 900 µL cells. For example, for a 100-ml transfection, 50–80 µg of DNA would be added to 10 mL of culture media, and then 130 µl of FectoPro would be added to this. After mixing and a 10-minute incubation, the resultant transfection cocktail would be added to 90 mL of cells. The cells were harvested 3–5 days after transfection by spinning the cultures at $>7000 \times g$ for 15 min. Supernatants were filtered using a 0.22-µm filter.

### Protein purification—Fc Tag-containing proteins

All proteins containing an Fc tag (for example, IgGs and hFc-ACE2) were purified using a 5-ml MabSelect SuRe PRISM column on the ÄKTA pure fast protein liquid chromatography (FPLC) system (Cytiva). Filtered cell supernatants were diluted with a 1/10 volume of 10× PBS. The ÄKTA system was equilibrated with: A1—1× PBS; A2—100 mM glycine pH 2.8; B1—0.5 M NaOH; Buffer line—1× PBS; and Sample lines—H$_2$O. The protocol washes the column with A1, followed by loading of the sample in Sample line 1 until air is detected in the air sensor of the sample pumps, followed by 5 column volume washes with A1 and elution of the sample by flowing of 20 ml of A2 (directly into a 50-ml conical containing 2 ml of 1 M Tris pH 8.0). The column was then washed with 5 column volumes of A1, B1, and A1. The resultant Fc-containing samples were concentrated using 50-kDa or 100-kDa cutoff centrifugal concentrators.

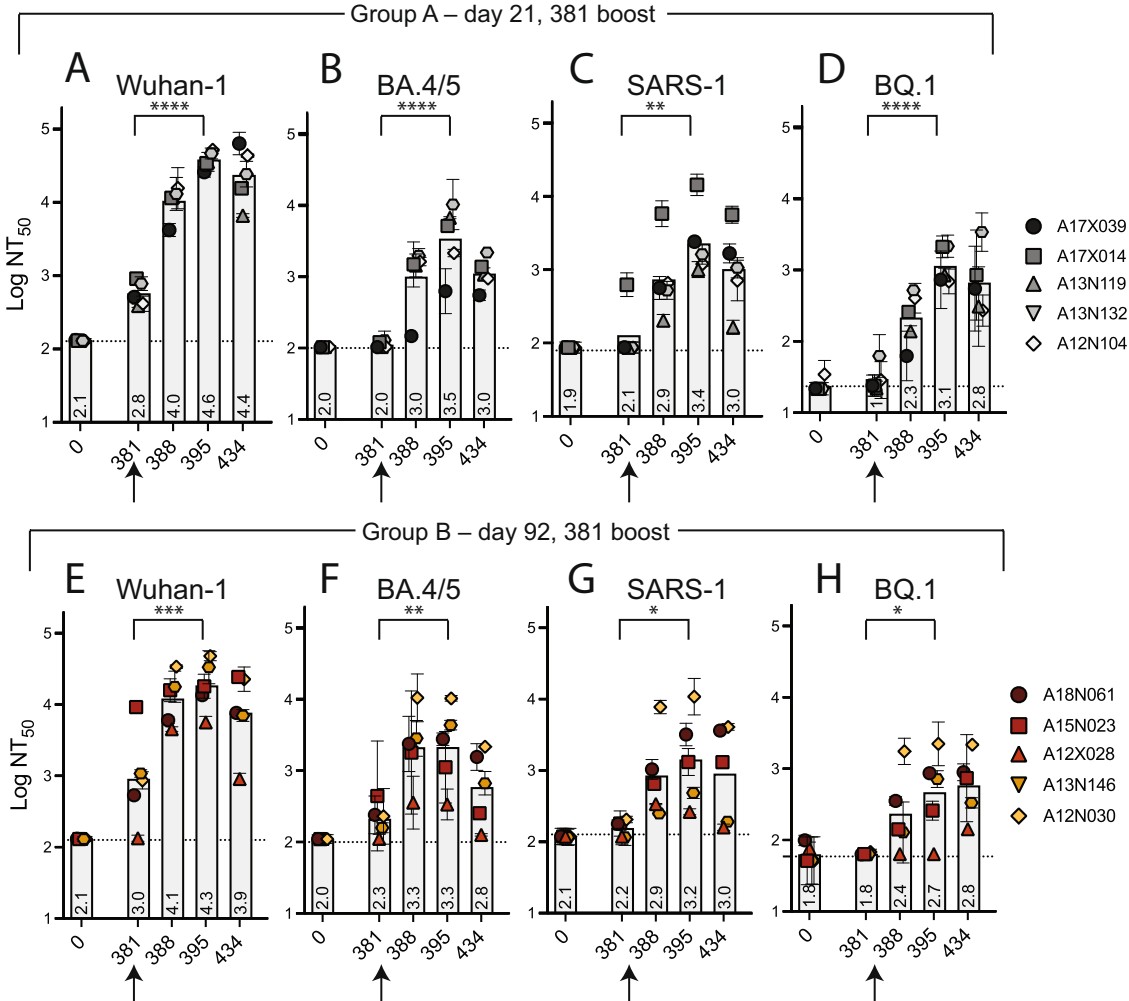

**Fig. 5 | Robust serum neutralizing anamnestic responses following a second booster of DCFHP-alum after ~one year in NHPs. A** Pseudovirus neutralization against Wuhan-1 (**A**), BA.4/5 (**B**), SARS-1 (**C**), or BQ.1 (**D**) by antisera from NHPs in group A following a boost at day 381. Days shown on x axis. **E**–**H** As in panels (**A**–**D**) but with NHPs in group B. NHP identification provided correlate with SI Table 1 ($n = 2$, $n =$ number of replicate neutralization assays conducted for these samples on independent days). Assay limits of quantitation indicated by horizontal dotted lines. Significance was tested between days 395 and days 381 using a nonparametric, one-way ANOVA comparing pre- and post-boost at day 381. ns = $P > 0.05$, *$P \leq 0.05$, **$P \leq 0.01$, ***$P \leq 0.001$, ****$P \leq 0.0001$. GMT (bars) and STD (for each animal) are shown. $P = < 0.0001$, <0.0001, 0.0013, <0.0001, 0.0006, 0.0043, 0.0298, 0.0232 for **A**–**H**, respectively.

## Protein purification–His-tagged proteins

Expi293F expressed SARS-CoV-2 RBD, containing a hexa-His tag was purified using HisPur Ni-NTA resin (Thermo Fisher Scientific). Expi cell supernatants were diluted with 1/3 volume of wash buffer (20 mM imidazole, 20 mM HEPES pH 7.4, and 150 mM NaCl), and the Ni-NTA resin was added to diluted cell supernatants and then samples were incubated at 4 °C while stirring overnight. Resin-supernatant mixtures were added to chromatography columns for gravity flow purification. The resin in the column was washed with 20 mM imidazole in 1xPBS, and the proteins were eluted with 250 mM imidazole in 1 x PBS. Column eluates were concentrated using centrifugal concentrators followed by size-exclusion chromatography on an ÄKTA pure system. ÄKTA pure FPLC with a Superose 6 Increase gel filtration column (S6) was used for further purification. One mL of sample was injected using a 2-ml loop and run over the S6, which had been pre-equilibrated in de-gassed 20 mM HEPES and 150 mM NaCl just before use.

## Nanoparticle purification–transient transfection

SΔC spike ferritin nanoparticles were isolated as previously described[14]. Briefly, they were purified using anion exchange chromatography, followed by size-exclusion chromatography using an SRT®

SEC-1000 column. Expi293F supernatants were concentrated using a AKTA Flux S column (Cytiva). The buffer was then exchanged with 20 mM Tris, pH 8.0 via overnight dialysis at 4 °C using 100 kDa molecular weight cut-off (MWCO) dialysis tubing. Dialyzed supernatants were filtered through a 0.22 μm filter and loaded onto a HiTrap® Q anion exchange column equilibrated in 20 mM Tris, pH 8.0. Spike nanoparticles were eluted using a 0–1 M NaCl gradient. Protein-containing fractions were pooled and concentrated using a 100 kDa MWCO Amicon® spin filter, and subsequently purified on a AKTA Pure system (Cytiva) using an SRT® SEC-1000 SEC column equilibrated in 1X PBS. Fractions were pooled based on $A_{280}$ signals and SDS-PAGE analysis on 4–20% Mini-PROTEAN® TGX™ protein gels stained with Gel-Code™ Blue Stain Reagent (ThermoFisher). Prior to immunizations or freezing, the samples were supplemented with 10% glycerol, filtered through a 0.22 μm filter, snap frozen, and stored at −20 °C until use.

DCFHP purification was done similar to above except via flow-through anion exchange followed by size-exclusion. Two buffers were initially prepared (buffer A: 20 mM Tris pH 8.0, buffer B: 20 mM Tris pH 8.0, 1 M NaCl). Filtered Expi293F or CHO supernatant was diluted with buffer B by 1/5 volume to a final concentration of 200 mM NaCl. The HiTrap® Q anion exchange column was washed with 5 column volumes (CV) sequentially with buffers A, B, A, prior to sample loading.

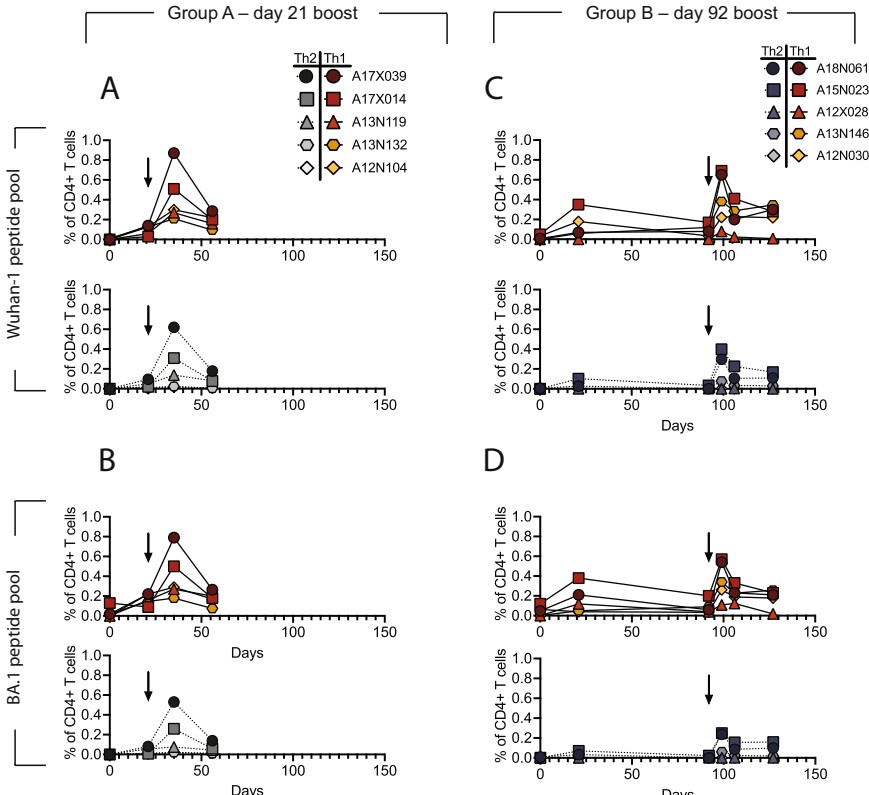

**Fig. 6 | DCFHP-alum immunized NHPs shows a balanced distribution of Th1 and Th2 CD4 + T cell responses.** T cells from animals in group A, isolated on the day shown on the x-axis, were stimulated with a peptide pool derived from Wuhan-1 (**A**) or Omicron BA.1 (**B**) spike protein and Th1 (top) and Th2 (bottom) cytokines were measured using flow cytometry (SI Fig. 9). Percent of CD4 + T cells that express either Th1 (IL-2, IFNg, or TNFa) (yellows and reds) or Th2 (IL-4) (grays) cytokines

following stimulation shows both are elicited following vaccination with DCFHP-alum. Arrow denotes day of boost. **C**, **D** As in panels (**A**) and (**B**) but with NHPs in group B. NHP identification provided correlate with SI Table 1. Percentages are shown as difference between treated samples and DMSO-treated control samples for a single experiment.

Diluted sample containing 200 mM NaCl was added to the column and the flow through was collected. One 5 mL HiTrap® Q anion exchange column was used for every 200 mL of diluted media. Multiple columns were joined in series for larger sample volumes. 100 kDa MWCO Amicon® spin filters were used to concentrate and buffer exchange the sample with 2 washes with 20 mM Tris, 150 mM NaCl pH 7.5. After the final wash, the sample was concentrated and filtered with a 0.22 μm filtered. The filtered sample was then loaded onto an SRT SEC-1000 column pre-equilibrated with 20 mM tris pH 7.5, 150 mM NaCl. The nanoparticle containing fractions were pooled as indicated in SI Fig. 1. Samples were routinely concentrated to 0.5–1 mg/mL and flash frozen in 20 mM Tris, 150 mM NaCl, 5% sucrose (weight:volume) buffer. Formulation with alum (Invivogen) was done by first extensively mixing (inversion 50 times) of the alum adjuvant and then mixing with appropriately diluted immunogen, at least 15 min prior to immunization.

## SDS-PAGE analysis
For the SDS-PAGE quantitation of DCFHP, samples diluted 10X in 1xPBS were mixed with the 2X reducing SDS sample buffer (900 μL 4x SDS PAGE sample buffer (Biorad) + 100 μL ß-mercaptoethanol + 2 ml of water) at 1:1 ratio and 10 μL of the mixture was loaded the 10-well 4–20% SDS-PAGE gradient gel. This loading is an equivalent to 0.5 μL of the starting media material. As a standard, each gel also included a mix of 1 μg of BSA + 1 μg of purified DCFHP from Expi293F cells. The quantitation was done with ImageJ software. The area of each band was determined, and the expression yield was calculated as Y = (Area of the band)/Area of the BSA standard*2 [g/L].

## Western blot analysis
Samples were diluted in SDS-PAGE Laemmli loading buffer (Bio-Rad) and electrophoresed on a 4–20% Mini-PROTEAN TGX protein gel (Bio-Rad). Proteins were transferred to nitrocellulose membranes using a Trans-Blot Turbo transfer system. Blots were blocked in 5% milk/PBST (1× PBS [pH 7.4], 0.1% Tween 20) and then washed with PBST. In-house-made primary antibody, CR3022 as previously described[14,70] (approximate concentration 0.8–1.3 mg/mL), was added at a 1:10,000 dilution in PBST. Blots were washed with PBST, and secondary rabbit anti-human IgG H&L HRP (abcam ab6759) was added at 1:10,000 in PBST. Blots were developed using Pierce ECL substrate and imaged using a GE Amersham imager 600.

## SEC-MALS of ferritin-antigen nanoparticles
SEC-MALS was performed on an Agilent 1260 Infinity II HPLC instrument with Wyatt detectors for light scattering (miniDAWN) and refractive index (Optilab). The purified antigen (20 μg) was loaded onto a SRT SEC-1000 4.6 mm × 300 mm column and equilibrated in 1× PBS (pH 7.4). SRT SEC-1000 column was run at a flow rate of 0.35 mL/min, and molecular weights were determined using ASTRA version 7.3.2 (Wyatt Technologies). For quantification, 75 μL of media sample was filtered with 0.22 um 96-well plate filters and 5 μL was injected into SEC SRT-1000 column. DCFHP expressed in Expi293F cells was used to generate a standard curve of 2.5, 5, 10, and 20 μg DCFHP protein. The area under peak (UV) for each curve was determined with Agilent software. The amount of the protein in the nanoparticle peak was interpolated from a standard curve using Prism interpolation function.

## BLI of mAbs binding to SARS-CoV-2 purified antigens

BLI was performed on an OctetRed 96 system (ForteBio). Samples were assayed in "Octet buffer" (0.5% bovine serum albumin, 0.02% Tween, 1×DPBS (1xDPBS from Gibco)) in 96-well flat-bottom black-wall, black-bottom plates (Greiner). Biosensors were equilibrated in Octet buffer for at least 10 min and regenerated in 100 mM glycine (pH 1.5) prior to sample testing. Tips in experiments that involved regeneration were regenerated in 100 mM glycine [pH 1.5] prior to testing. Anti-Human Fc sensor tips (ForteBio, now Sartorius) were loaded with 200 nM mAb at a threshold of 0.8 nm and then submerged into wells containing 100 nM (protomer/monomer concentration) of each antigen. For samples where BLI was being used to quantitate concentrations, a standard curve was made using purified DCFHP in PBS, where the final association signal used to determine the standard curve.

Specifically, COVA 2-15 antibody was used for standard curve development and tested on day 13 media samples of stably expressing CHO cells. Media was diluted to a final dilution of 1:200. COVA2-15 antibody was diluted to 100 nM concentration in octet buffer and loaded on Fc-binding octet tip to threshold of 0.4 The tips was then moved into octet buffer for 30 s and then into well containing the 200X diluted media samples to obtain the end-point readout after 30 s of association (readout averaged at 29.5–29.9 s). The tips were regenerated, as above, twice before starting the experiment. The yield was calculated using interpolation from the standard curve determined with DCFHP using Prism Version 9.4.1.

## Differential scanning fluorimetry

Thermal melts were determined using the Prometheus NT.48 made by Nanotemper. Samples were loaded into Prometheus NT.Plex nanoDSF Grade High Sensitivity Capillary Chips and the laser intensity was set such that the discovery scan placed the auto fluorescence between the upper and lower bounds. Samples were let to melt using the standard melt program (1 °C/min). Melting temperatures were determined by peaks on the first derivatives of the ratio of F350/F330.

## Stable cell line production

Stable cell line production was completed at ATUM (Newark CA) under GMP conditions as described previously[47,48]. Briefly, DNA sequences encoding DCFHP were chemically synthesized and transformed into *E. coli* hosts. In addition to DCFHP, the constructs were designed to express glutamine synthetase. Sequence confirmation of the DCFHP gene and whole plasmid sequence was routinely conducted. In addition, Leap-In Transposase system ORF expression constructs were designed and constructed based on ATUM's proprietary technology. *E. coli* hosts containing DCFHP genes were maxi-prepped (Macherey-Nagel). HD-BIOP3 cells were then transfected with both components and underwent selection by glutamine deprivation[71]. Resulting pooled cells were sorted and resulting single clones were analyzed as described throughout the manuscript. Optimal clones were selected for cell banking.

## Lentivirus production

SARS-CoV-2 VOCs and SARS-CoV-1 spike pseudotyped lentiviral particles were produced. HEK293T cells were transfected with plasmids described above for pseudoviral production using BioT transfection reagent. Six million cells were seeded in D10 media (DMEM + additives: 10% FBS, L-glutamate, penicillin, streptomycin, and 10 mM HEPES) in 10-cm plates 1 day before transfection. A five-plasmid system (plasmids described above) was used for viral production, as described in Crawford et al.[46]. The spike vector contained the 21-amino acid truncated form of the SARS-CoV-2 spike sequence from the Wuhan-Hu-1 strain of SARS-CoV-2 or VOCs or 18-amino acid truncation for SARS-CoV-1. VOCs were based on wild-type (WT) (Uniprot ID: BCN86353.1); Alpha (sequence ID: QXN08428.1); Beta (sequence ID: QUT64557.1); Gamma (sequence ID: QTN71704.1); Delta (sequence ID: QWS06686.1,

which also has V70F and A222V mutations); and Omicron (sequence ID: UFO69279.1) specific mutations shown in SI Table 3. The plasmids were added to D10 medium in the following ratios: 10 μg pHAGE-Luc2-IRS-ZsGreen, 3.4 μg FL spike, 2.2 μg HDM-Hgpm2, 2.2 μg HDM-Tat1b and 2.2 μg pRC-CMV-Rev1b in a final volume of 1000 μl. To form transfection complexes, 30 μl of BioT (BioLand) was added. Transfection reactions were incubated for 10 min at room temperature, and 9 ml of medium was added slowly. The resultant 10 ml was added to plated HEK cells from which the medium had been removed. Culture medium was removed 24 h after transfection and replaced with fresh D10 medium. Viral supernatants were harvested 72 h after transfection by spinning at $300 \times g$ for 5 min, followed by filtering through a 0.45-μm filter. Viral stocks were aliquoted and stored at −80 °C until further use.

## Neutralization

The target cells used for infection in viral neutralization assays were from a HeLa cell line stably overexpressing the SARS-CoV-2 receptor, ACE2, as well as the protease known to process SARS-CoV-2, TMPRSS2. Production of this cell line is described in detail in ref. [72] with the addition of stable TMPRSS2 incorporation. ACE2/TMPRSS2/HeLa cells were plated 1 day before infection at 10,000 cells per well. Ninety-six-well white-walled, clear-bottom plates were used for the assay (Thermo Fisher Scientific) and a white seal was placed on the bottom prior to readout. As previously described[14], on the day of the assay, dilutions of heat inactivated serum were made into sterile D10 medium. Samples were analyzed in technical duplicate in each experiment. Virus-only wells and cell-only wells were included in each assay. In addition, a COVA2-15 positive neutralization control was included in each assay to confirm replicability of the experiment.

A virus dilution was made containing the virus of interest (for example, SARS-CoV-2) and D10 media (DMEM + additives: 10% FBS, L-glutamate, penicillin, streptomycin, and 10 mM HEPES). Virus dilutions into media were selected such that a suitable signal (>1,000,000 RLU) would be obtained in the virus-only wells. Polybrene was present at a final concentration of 5 μg/mL in all samples. 50 μL of heat inactivated sera was mixed with 50 μL viral dilution to make a final volume of 120 μl In each well.

The inhibitor (serum dilution) and virus mixture was left to incubate for 1 h at 37 °C. After incubation, the medium was removed from the cells on the plates made 1 day prior. This was replaced with 100 μl of inhibitor/virus dilutions and incubated at 37 °C for approximately 48 h. Infectivity readout was performed by measuring luciferase levels. 48 h after infection, the medium was removed from all wells and cells were lysed by the addition of 100 μl of a 1:1 dilution of BriteLite assay readout solution (Perkin Elmer) and 1xDPBS (Gibco) into each well. Luminescence values were measured using a BioTek Synergy HT (BioTek) or Tecan M200 microplate reader. Each plate was normalized by averaging cell-only (0% infectivity) and virus-only (100% infectivity) wells. Cell-only and virus-only wells were averaged. Normalized values were fit with a four-parameter non-linear regression inhibitor curve in Prism to obtain 50% neutralizing titer (NT₅₀) values. Half-life values were calculated starting from the maximum serum titer obtained for each sample. Half-life values were first calculated via a 1 phase decay, which, when unconstrained, demonstrated high plateau values of serum neutralization around $10^3$, demonstrating the importance of constraining the plateau value to close to 0. Constraining the plateau to $\log(10^0)$, we tested both one-phase (monophasic) and two-phase (biphasic) decay. The fast phase of the two-phase decay fit well for all animals (SI Table 5) but given the slow decline of neutralizing titers, there was not enough data to fully fit the slow phase decay for some animals, specifically those in group B. in such cases where estimated half-life was >5000 days, 5000 days was used for average calculations shown in SI Table 5. LOQ was set as the neutralizing titer of day 0 serum or the lowest serum

dilution tested, whichever was higher. For samples with different LOQs on the same graph, the average value was used.

## ELISA

RBD (5 μg/mL) was plated in 50 μl in each well on a MaxiSorp (Thermo Fisher Scientific) microtiter plate in 1xPBS and left to incubate for at least 1 h at room temperature. These were washed 3 times with 300 μl of ddH$_2$O using an ELx 405 Bio-Tex plate washer and blocked with 150 μl of ChonBlock (Chondrex) for at least 1 h at room temperature. Plates were washed 3x with 300 μl of 1x PBST. Mouse serum samples, serially diluted in diluent buffer (1x PBS, 0.1% Tween) starting at 1:50 serum dilution was then added to coated plates for 1 h at room temperature. This was then washed 3x with PBST. HRP goat anti-mouse (BioLegend 405306) was added at a 1:10,000 dilution in diluent buffer for 1 h at room temperature. This was left to incubate at room temperature for 1 h and then washed 6x with PBST. Finally, the plate was developed using 50 μl of 1-StepTM Turbo-TMB-ELISA Substrate Solution (Thermo Fisher Scientific) per well, and the plates were quenched with 50 μl of 2 M H$_2$SO$_4$ to each well. Plates were read at 450 nm and normalized for path length using a BioTek Synergy HT Microplate Reader.

## Live SARS-CoV-2 virus isolation and passages

Variants were obtained from two sources. WA-1/2020 was obtained from the WRCEVA collection. BA.1 and BA.2 were isolated from de-identified nasopharyngeal (NP) swabs sent to the California Department of Public Health from hospitals in California for surveillance purposes. To isolate from patient swabs, 200 μl of an NP swab sample from a COVID-19-positive patient that was previously sequence-identified was diluted 1:3 in PBS supplemented with 0.75% BSA (BSA-PBS) and added to confluent Vero E6-TMPRSS2-T2A-ACE2 cells in a T25 flask, allowed to adsorb for 1 h, inoculum removed, and additional media was added. The flask was incubated at 37 °C with 5% CO$_2$ for 3–4 days with daily monitoring for cytopathic effects (CPE). When 50% CPE was detected, the contents were collected, clarified by centrifugation and stored at −80 °C as a passage 0 stock. 1:10 diluted passage 0 stock was used to inoculate Vero E6-TMPRSS2-T2A-ACE2 grown to confluency in T150 flasks and harvested at approximately 80% CPE. All viral stocks were sequenced to confirm lineage, and 50% tissue culture infectious dose (TCID$_{50}$) was determined by titration.

## Live SARS-CoV-2 virus 50% CPE endpoint neutralization

CPE endpoint neutralization assays were performed following the limiting dilution model using sequence-verified viral stocks of WA-1, BA.1, and BA.2 in Vero E6-TMPRSS2-T2A-ACE2. Three-fold serial dilutions of inhibitor (antisera) were made in BSA-PBS and mixed at a 1:1 ratio with 100 TCID$_{50}$ of each virus and incubated for 1 h at 37 °C. Final inhibitor dilutions ranged from 500 nM to 0.223 nM. Then, 100 μl of the plasma/virus mixtures were added in duplicate to flat-bottom 96-well plates seeded with Vero E6-TMPRSS2-T2A-ACE2 at a density of $2.5 \times 10^4$ per well and incubated in a 37 °C incubator with 5% CO$_2$ until consistent CPE was seen in the virus control (no inhibitor added) wells. Positive and negative controls were included as well as cell control wells and a viral back titration to verify TCID$_{50}$ viral input. Individual wells were scored for CPE as having a binary outcome of 'infection' or 'no infection', and the IC$_{50}$ was calculated using the Spearman–Karber method. All steps were done in a Biosafety Level 3 laboratory using approved protocols.

## Cryo-EM data acquisition

Previously frozen DCFHP protein was quick-thawed at room temperature and subjected for gel filtration and concentrated to 0.4 mg/mL in a buffer of 150 mM NaCl, 20 mM HEPES pH 7.4. 3 μL of the DCFHP proteins were applied onto a glow-discharged Quantifoil R 1.2/1.3 Cu 300-mesh grid (Quantifoil). The grids were blotted for 2 s at

20 °C and 100% humidity and rapidly cryocooled in liquid ethane using a Vitrobot Mark IV instrument (Thermo Fisher Scientific). The DCFHP proteins were imaged at 0.86 Å per pixel by a Titan Krios cryo-electronmicroscope, TEM Beta (Thermo Fisher Scientific) at Stanford-SLAC Cryo-EM CenterS2C2, operated at 300 kV. Micrographs were recorded with EPU (Thermo Fisher Scientific) with a Gatan K2 Summit direct electron detector. Each movie was composed of 40 frames with an exposure time of 2.4 s and 50.187 electron dose. A data set of 8750 movie stacks was collected.

## Single-particle image processing and 3D reconstruction

All 8750 movies were imported into cryoSPARC 3.2[73]. Motion correction was performed in patch motion correction and the contrast transfer function (CTF) was determined in patch CTF estimation[74]. 601 DCFHP single particles were manually picked from 93 micrographs in Manual Picker. Four 2D classes were determined in 2D classification, which were used as templates in Template picker and a total of 1,590,688 particles was picked from 8750 movies.

By 2D classification, 127,630 particles of DCHPF were selected and used for Ab-initio reconstruction and homogeneous refinement with C1 symmetry, the CryoEM density of the ferritin core was well resolved, while that of the Spike trimer are not, suggesting that the conformational orientation of the Spike trimer displayed on the 3-axis symmetry of ferritin is flexible. Through iterative 2D classifications, 67,324 particles showing relatively homogenous Spike density from five 2D classes were used for generating an initial model in Ab-initio reconstruction. All eight spike trimers were resolved by homogeneous refinement enforced with octahedral symmetry. A Gaussian low-pass filter was applied to the CryoEM maps displayed in UCSF Chimera[75].

## Mouse immunizations

Balb/c female mice (6–8 weeks old) were purchased from The Jackson Laboratory. All mice were maintained at Stanford University according to the Public Health Service Policy for "Humane Care and Use of Laboratory Animals" following a protocol approved by Stanford University Administrative Panel on Laboratory Animal Care (APLAC-33709). Mice were immunized intramuscularly with antigen doses indicated in figure legends. With the exception of the high dose alum and alum/CpG (Fig. 1 and SI Figs. 1 and 2 which contain 500 μg alum ± 20 μg CpG), all antigen doses were formulated with 150 μg alum in Tris Buffer (20 mM, pH 7.5, 150 mM NaCl, 5% sucrose) in a total volume of 100 μL per injection. DCFHP was produced in Expi293F cells via transient transfection for all mouse experiments excluding the temperature stability experiment where the sample was produced in the described CHO cell-line. Mice immunization schedules were as described in the Figure legends. Serum was collected and processed using Sarstedt serum collection tubes. Mouse serum was centrifuged at 10,000 × g for 15 min and heat inactivated for 30 min at 56 °C.

## NHP studies

Ten male rhesus macaques (Macaca mulatta) of Indian origin, aged 3–9 years, were assigned to the study (SI Table 1). The animals were distributed between the two groups such that the age and weight distribution were comparable across them. Animals were housed and maintained at the New Iberia Research Center (NIRC) of the University of Louisiana at Lafayette in accordance with the rules and regulations of the Committee on the Care and Use of Laboratory Animal Resources. The entire study (IACUC approval number: 2021-012-8738) was reviewed and approved by the University of Louisiana at Lafayette Institutional Animal Care and Use Committee (IACUC) and Stanford University APLAC committee (Protocol # 34139). All animals were negative for simian immunodeficiency virus, simian T cell leukemia virus and simian retrovirus. Animals were immunized according to the schedule outlined in SI Table 2. DCFHP for NHP vaccines was isolated from a pool of the top 24 expressing CHO cell lines.

### Intracellular cytokine staining assay

Antigen-specific T cell responses were measured using the intracellular cytokine staining assay as previously described[52]. Live frozen PBMCs were thawed, counted, and resuspended at a density of $10^6$ live cells/mL in complete RPMI (RPMI supplemented with 10% FBS and antibiotics). The cells were rested overnight at 37 °C in a $CO_2$ incubator. The following day the cells were counted again, resuspended at a density of $15 \times 10^6$ cells/mL in complete RPMI and 100 μL of cell suspension containing $1.5 \times 10^6$ cells was added to each well of a 96-well round-bottomed tissue culture plate. Each cell sample was treated with three conditions: no stimulation (DMSO), a peptide pool spanning the spike protein at a concentration of 1.2 μg/mL of each peptide, and a peptide pool spanning the spike protein of Omicron BA.1 (1.2 μg/mL of each peptide). This was done in the presence of 1 μg/mL of anti-CD28 (clone CD28.2, BD Biosciences) and anti-CD49d (clone 9F10, BD Biosciences) as well as anti-CXCR3 and anti-CXCR5. The peptides were custom-synthesized to 90% purity using GenScript, a commercial vendor. All samples contained 0.5% (v/v) DMSO in total volume of 200 μL per well. The samples were incubated at 37 °C in $CO_2$ incubators for 2 h before addition of 10 μg/mL brefeldin A. The cells were incubated for an additional 4 h. The cells were washed with PBS and stained with Zombie UV fixable viability dye (Biolegend). The cells were washed with PBS containing 5% FCS, before the addition of surface antibody cocktail. The cells were stained for 20 min at 4 °C in 100 μL volume. Subsequently, the cells were washed, fixed, and permeabilized with cytofix/cytoperm buffer (BD Biosciences) for 20 min. The permeabilized cells were stained with intracellular cytokine staining antibodies for 20 min at room temperature in 1× perm/wash buffer (BD Biosciences). Cells were then washed twice with perm/wash buffer and once with staining buffer before acquisition using the BD Symphony Flow Cytometer and the associated BD FACS Diva software. All flow cytometry data were analyzed using Flowjo software v10 (TreeStar Inc.).

### Reporting summary

Further information on research design is available in the Nature Portfolio Reporting Summary linked to this article.

## Data availability

Sequences are described or outlined in SI tables and raw data are plotted as shown or included as tables. Raw data for the figures is included in the source data document. SARS-CoV-2 spike protein sequences were taken from GISAID as outlined in SI Table 3. Source data are provided with this paper.

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

## Acknowledgements

We thank members of the Kim Lab for discussions, and Drs. David Baltimore, Luciana Borio, and Duo Xu for helpful comments on an earlier draft of this manuscript. S.T. acknowledges NIH NICHD grant K99HD104924 and the Merck fellowship from the Damon Runyon Cancer Research Foundation. The Cryo-EM microscopy work was performed at the Stanford-SLAC Cryo-EM Center (S2C2) supported by the NIH Common Fund Transformative High Resolution Cryo-Electron Microscopy program (U24 GM129541). We thank F. Krammer and F. Amanat for providing the SARS-CoV-2 RBD plasmids for protein production, and J. Bloom and A. Greaney for plasmids and cells related to viral neutralization assays. The findings and conclusions in this article are those of the author(s) and do not necessarily represent the views or opinions of the California Department of Public Health or the California Health and Human Services Agency. This work was supported by the Frank Quattrone & Denise Foderaro Family Research Fund, the Chan Zuckerberg Biohub, the Stanford Innovative Medicines Accelerator, the Virginia & D.K. Ludwig Fund for Cancer Research, and an NIH Director's Pioneer Award (DP1AI158125) to P.S.K. A previous version of this manuscript appeared on bioRxiv (https://doi.org/10.1101/2022.12.25.521784).

## Author contributions

P.A.-B.W., M.S., and N.F. contributed equally to this work. P.A.-B.W. was responsible for construct design, initial characterization, and assay development. M.S. was responsible for mouse immunizations and lentiviral neutralization assays. N.F. was responsible for protein purification and characterization as well as cell clone evaluation. S.T. conducted cryo-EM analysis. P.S.A. and M.H. were responsible for T cell experiments. O.S.K. contributed to protein characterization and assay development. M.K.M. conducted live viral neutralization assays. J.F. and L.S. were involved in NHP immunization and sera collection. J.D. contributed to mouse immunizations and lentiviral neutralization assays. Y.C.C. contributed to cell clone evaluation and lentiviral neutralization assays. G.V. and M.B.F. advised on cell line development. F.J.V. supervised NHP work. C.H. supervised live viral neutralization work. S.B.J. and D.B.V. advised on assay development and product characterization. B.P. supervised T cell experiments and advised on immunoassays. P.S.K. supervised the project. P.A.-B.W. developed figures and P.A.-B.W. and P.S.K. wrote the manuscript.

## Competing interests

P.A.B.W., M.S., N.F., S.T., and P.S.K. are named as inventors on patent applications applied for by Stanford University and the Chan Zuckerberg Biohub on immunogenic coronavirus fusion proteins and related methods, which have been licensed to Vaccine Company, Inc. P.A.B.W. is an employee of, and P.S.K. is a co-founder and member of the Board of Directors of Vaccine Company, Inc. All other authors declare no competing interests.
