## [Peer Review File · Nature Communications]

A ferritin-based COVID-19 nanoparticle vaccine that elicits robust, durable, broad-spectrum neutralizing antisera in non-human primatesReviewers' Comments:

Reviewer #1:

Remarks to the Author:

In this manuscript, the authors describe a ferritin-based nanoparticles vaccine candidate and demonstrated its immunogenicity in NHP. Although similar nanoparticle vaccine candidates have been reported before, this study shows that using a 6P stabilised Spike in the ferritin nanoparticle improves the yield while retaining equal immunogenicity compared to the 2P version. Nanoparticles vaccines have become popular in recent years and their superiority over simple soluble antigens become increasingly clear over time. As shown in this study, they managed to achieve >2g/L of culture which is impressive. They have also shown that, when coupled to alum, the vaccine candidate remains stable at 37C for at least 14 days which means a cold-chain might not be required to distribute the vaccine. The study also shows that a booster dose after ~1 year and restore the neutralising antibody responses showing the potential of the vaccine to be used as a booster vaccine in vaccinated humans.

Comments:

1. Similar nanoparticle vaccine has been reported previously (Joyce et.al., Sci Trans Med, 2021 and Carmen et.al., Vaccines, 2021). Thus, the novelty of the vaccine approach for SARS-CoV-2 pre-fusion stabilised Spike ferritin nanoparticle vaccine is not high although some incremental progress is presented (e.g a 6P stabilised version is used instead of 2P)
2. Please state precisely in the manuscript which type of alum adjuvant was used in the study. Was it aluminium hydroxide?
3. Statistical analysis is missing in Figure 3, 4, 5 and 6.
4. Figure 2. Have you compared the stability of SΔC Fer to DCFHP? Not essential for this manuscript but it's interesting to know if the 6P offers increased thermostability.
5. In Figure 2. Apart from immunogenicity. Have any characterisations on the alum:DCFHP formulation been done? E.g. the preservation of the folding state of the nanoparticle absorbed on the alum gel?
6. Figure 3. I am a bit confused about what the authors called "biological replicate". For instance, in Figure 3C, it says in the figure legend that "means and SD for each biological replicates are plotted and noted for each animal" with an "n" stated below the graph. I would assume that "biological replicate" mean the numbers of NHP (which is 5 in this case) and in that case does "n" actually mean "technical replicate"? Please clarify.
7. Line 163. "Sarbecovirus which stands for SARS-like betacoronavirus" instead of "betasarbecovirus".
8. SI Table 2. I think the unit for Dose and Alhydrogel should be "μg" instead of "mg"

Reviewer #2:

Remarks to the Author:

Payton A.-B. Weidenbacher and colleagues presented the data on the design and characterization of an improved ferritin-based nanoparticle vaccine, SΔC-Fer Powell AE. et al., ACS Cent Sci. 2021) called DCFHP (Delta-C70-Ferritin-HexaPro). This improved vaccine contains the deletion of 70 C-terminal spike ectodomain residues (SΔC-Fer), and also incorporates four proline residue substitutions and a modification to the mutated furin cleavage site to stabilize and promote robust expression. The DCFHP is more stable to thermal denaturation than SΔC-Fer. In addition, the improved vaccine contains alum (Alhydrogel™ adjuvant) to create a DCFHP-alum formulation. This design, removal of C-terminal immunodominant linear epitopes, and multivalent presentation of the modified spike protein on a ferritin nanoparticle substantially improved the immunogenicity and elicited a robust and durable immune response in mice, and a two-dose intramuscular immunization regimen in rhesus macaques with DCFHP-alum elicits antisera with durable, robust and broad neutralization of VOCs, including the Omicron subvariants BA.4/537 and BQ.1,38 along with a balanced Th1 and Th2 immune response. Further, boosting these immunized NHPs after ~1 year with a third dose of DCFHP-alum elicited a

robust, broad-spectrum, anamnestic neutralizing antibody response. The authors have analyzed the neutralization titers extensively (most measurements are with pseudovirus neutralization assay); however, several drawbacks are reducing the enthusiasm in the manuscript. These include:

- i) While authors compared Joyce, M. G. et al. *Sci Transl Med* 2022, and Wuertz, K. M. et al. *NPJ Vaccines* 2021 studies to show neutralization data with the protection in hamster and NHPs, the lack of protection data is the major setback of the study.
- ii) One of the strengths of the manuscript includes the antigen design. However, the physical characterization shows increased stability and expression levels but not translated into increased neutralization antibody responses. For instance, several recent studies showed that the furin cleavage inactivation in the background of two proline mutations (2-P) significantly enhanced neutralization (assessed using live virus) titers.
- iii) Fig 4 – NHPs immunized with DCFHP-alum elicited long-lived immunity against both Wuhan-1 and BA.4/5 pseudoviruses, however the serum neutralizing titers against Wuhan-1 in group A appears to be always higher (although authors have not discussed the absolute values in the text or presented on the graphs) compared to group B, authors need to present the peak data in the separate bar graphs and perform significance test. Also, explicitly state the fold differences between groups A and B.
- iv) Similarly, Fig 5 – Serum neutralizing antibody titers following a second booster of DCFHP-alum after ~one year (while both the groups were given the vaccine at day 381), the pseudovirus neutralization assay against Wuhan-1, and VOCs BA.4/5 and BQ.1, show similar titers in both the groups, I don't really see the significance of delayed 2nd dose.
- v) Fig 6 – There are only CD4 T cell responses, presented only until two diseases (although the third dose data is there for antibodies, not for the T cell responses) shows a balanced distribution of Th1 and Th2 CD4+ T cell responses, it is not anything further to the manuscript.
- vi) One of the current challenges for the COVID-19 vaccines is immune responses in the mucosal immune response, and the authors did not show any evidence of the mucosal immune response with the current vaccine.
- vii) Also, the selection of macaques is poorly stated, as the vaccine targets pediatric populations, but the study was conducted using male rhesus macaques aged between 3 and 9 years. Authors need to explicitly state the age of the monkey and the age of the humans in their NHP study design and the text.

Overall, the manuscript contains some important findings however, the lack of protection against current emerging VOCs following vaccination is a major setback for the manuscript.

REVIEWER COMMENTS

Reviewer #1 (Remarks to the Author):

In this manuscript, the authors describe a ferritin-based nanoparticles vaccine candidate and demonstrated its' immunogenicity in NHP. Although similar nanoparticle vaccine candidates have been reported before, this study shows that using a 6P stabilised Spike in the ferritin nanoparticle improves the yield while retaining equal immunogenicity compared to the 2P version. Nanoparticles vaccines have become popular in recent years and their superiority over simple soluble antigens become increasingly clear over time. As shown in this study, they managed to achieve >2g/L of culture which is impressive. They have also shown that, when coupled to alum, the vaccine candidate remains stable at 37C for at least 14 days which means a cold-chain might not be required to distribute the vaccine. The study also shows that a booster dose after ~1 year and restore the neutralising antibody responses showing the potential of the vaccine to be used as a booster vaccine in vaccinated humans.

We thank the reviewer for their comments and considerations. We agree that the results are impressive and demonstrate improvements overs previous vaccines.

Comments:

1. Similar nanoparticle vaccine has been reported previously (Joyce et.al., Sci Trans Med, 2021 and Carmen et.al., Vaccines, 2021). Thus, the novelty of the vaccine approach for SARS-CoV-2 pre-fusion stabilised Spike ferritin nanoparticle vaccine is not high although some incremental progress is presented (e.g a 6P stabilised version is used instead of 2P)

We agree with this discussion and have added the suggested citations and highlighted the differences in the main text (pg 3):

“We supplemented the 2P stabilizing substitutions with four previously described (26) proline substitutions to create a six-proline substituted (HexaPro) version of the vaccine. Previous work has shown that the HexaPro SARS-CoV-2 spike protein has increased stability and expression relative to the 2P version (26) further differentiating this vaccine from previous spike-ferritin products (16,30).“

2. Please state precisely in the manuscript which type of alum adjuvant was used in the study. Was it aluminium hydroxide?

We agree this is an important designation. The main text includes the text (pg 3):

“Our vaccine formulation, DCFHP-alum, consists of DCFHP antigen formulated with aluminum hydroxide (Alhydrogel™, referred to herein as alum) as the only adjuvant“

3. Statistical analysis is missing in Figure 3, 4, 5 and 6.

We have included the relevant statistical analysis for Figures 3, in a new SI figure (SI Fig 5), as per the recommendation of reviewer 2. Additionally, we've added statistical analysis in figure 5. We hesitate to add statistical analysis to Figures 4 and 6 as these are intended to represent the trends of data at whole as opposed to make specific statistical claims about the significance.

4. Figure 2. Have you compared the stability of Δ C Fer to DCFHP? Not essential for this manuscript but it's interesting to know if the 6P offers increased thermostability.

We have included thermal stability in SI Figure 1.

5. In Figure 2. Apart from immunogenicity. Have any characterisations on the alum:DCFHP formulation been done? E.g. the preservation of the folding state of the nanoparticle absorbed on the alum gel?

Thank you for this question. This has been studied in depth in a partner manuscript in development now and cited within the manuscript.

6. Figure 3. I am a bit confused about what the authors called "biological replicate". For instance, in Figure 3C, it says in the figure legend that "means and SD for each biological replicates are plotted and noted for each animal" with an "n" stated below the graph. I would assume that "biological replicate" mean the numbers of NHP (which is 5 in this case) and in that case does "n" actually mean "technical replicate"? Please clarify.

We thank the reviewer for highlighting this confusion. The biological replicates we define as "n" are not the NHPs, but rather are replicates of the neutralization experiments used to determine the NT50. The replicates are conducted on different days and often by different scientists. We have updated the text in the legends for figures 3, 4 and 5 accordingly to clarify this point (pgs 10, 11, 13):

"n = number of replicate neutralization assays conducted for these samples on independent days"

7. Line 163. "Sarbecovirus which stands for SARS-like betacoronavirus" instead of "betasarbecovirus".

We thank the reviewer for identifying this mistake and have updated the text accordingly (pg. 9).

8. SI Table 2. I think the unit for Dose and Alhydrogel should be " μ g" instead of "mg"

We thank the reviewer for identifying this mistake and have updated the text accordingly.

Reviewer #2 (Remarks to the Author):

Payton A.-B. Weidenbacher and colleagues presented the data on the design and characterization of an improved ferritin-based nanoparticle vaccine, Δ C-Fer Powell AE. et al., ACS Cent Sci. 2021) called DCFHP (Delta-C70-Ferritin-HexaPro). This improved vaccine contains the deletion of 70 C-terminal spike ectodomain residues (Δ C-Fer), and also incorporates four proline residue substitutions and a modification to the mutated furin cleavage site to stabilize and promote robust expression. The DCFHP is more stable to thermal denaturation than Δ C-Fer. In addition, the improved vaccine contains alum (Alhydrogel™ adjuvant) to create a DCFHP-alum formulation. This design, removal of C-terminal immunodominant linear epitopes, and multivalent presentation of the modified spike protein on a ferritin nanoparticle substantially improved the immunogenicity and elicited a robust and durable immune response in mice, and a two-dose intramuscular immunization regimen in rhesus macaques with DCFHP-alum elicits antisera with durable, robust and broad neutralization of VOCs, including the Omicron subvariants BA.4/537 and BQ.1,38 along with a balanced Th1 and Th2 immune response.

Further, boosting these immunized NHPs after ~1 year with a third dose of DCFHP-alum elicited a robust, broad-spectrum, anamnestic neutralizing antibody response. The authors have analyzed the neutralization titers extensively (most measurements are with pseudovirus neutralization assay); however, several drawbacks are reducing the enthusiasm in the manuscript. These include:

We thank the reviewer for their time reviewing the manuscript and appreciate their feedback.

i) While authors compared Joyce, M. G. et al. *Sci Transl Med* 2022, and Wuertz, K. M. et al. *NPJ Vaccines* 2021 studies to show neutralization data with the protection in hamster and NHPs, the lack of protection data is the major setback of the study.

While we agree that a challenge study in a hamster model would show further evidence that this vaccine is conferring robust protection, the models are limited on their ability to test Omicron subvariants and neutralizing titers are an established surrogate. We reference the papers that establish this surrogate relationship:

“In addition, clinical trials have established a correlation between anti-SARS-CoV-2 monoclonal antibody levels (i.e., humoral immunity alone) and protection from COVID-19.^{60,61} Indeed, SARS-CoV-2 variant booster vaccines have been accepted by the FDA and EMA for emergency use authorization using neutralizing antibody titer as a correlate of protection.^{62,63}” (pg. 15).

As such, we envision this experiment to be outside the scope of the current manuscript.

ii) One of the strengths of the manuscript includes the antigen design. However, the physical characterization shows increased stability and expression levels but not translated into increased neutralization antibody responses. For instance, several recent studies showed that the furin cleavage inactivation in the background of two proline mutations (2-P) significantly enhanced neutralization (assessed using live virus) titers.

This is an interesting comment and we have added the importance of inactivation of the polybasic site to the introduction and included a citation (pg. 2):

“ Δ C-Fer contains an inactivated polybasic cleavage site, which has been shown to improve neutralizing titers,²³ and the 2-proline (2P)²⁴ prefusion-stabilizing substitutions found in the FDA-approved SARS-CoV-2 mRNA vaccines^{1,25}.”

Amanat, F. et al. Introduction of Two Prolines and Removal of the Polybasic Cleavage Site Lead to Higher Efficacy of a Recombinant Spike-Based SARS-CoV-2 Vaccine in the Mouse Model. *mBio* 12 (2021). <https://doi.org:10.1128/mBio.02648-20>

iii) Fig 4 – NHPs immunized with DCFHP-alum elicited long-lived immunity against both Wuhan-1 and BA.4/5 pseudoviruses, however the serum neutralizing titers against Wuhan-1 in group A appears to be always higher (although authors have not discussed the absolute values in the text or presented on the graphs) compared to group B, authors need to present the peak data in the separate bar graphs and perform significance test. Also, explicitly state the fold differences between groups A and B.

This is an interesting suggestion, and one we had not considered this given the improved variant response seen in group B. We have performed the statistical analysis suggested and included it in an

additional supplemental figure 5. This analysis revealed that, indeed group A shows significantly stronger responses against Wuhan-1 – but significantly weaker responses against the omicron variants. No significance was seen against SARS-1. These results suggest that more is at play in the delayed boost than simply robust neutralizing titers.

We have added a description of this figure to the main text (pg. 9):

“Interestingly, while group A elicited a more robust response against Wuhan-1 (SI Fig 5), on average, group B showed approximately a 4-fold increased neutralizing response relative to group A against divergent VOCs (SI Fig 5).”

iv) Similarly, Fig 5 – Serum neutralizing antibody titers following a second booster of DCFHP-alum after ~one year (while both the groups were given the vaccine at day 381), the pseudovirus neutralization assay against Wuhan-1, and VOCs BA.4/5 and BQ.1, show similar titers in both the groups, I don't really see the significance of delayed 2nd dose.

We agree that the delayed 2nd dose shows the most robust difference in the first year window, but following an additional delayed dose for all groups they are more similar. We envision this is essentially because both groups have now experienced a delayed boost. We have highlighted this more explicitly in the text (pg. 12):

“Boosting after one year essentially abrogated the differences seen between groups A and B.”

v) Fig 6 – There are only CD4 T cell responses, presented only until two diseases (although the third dose data is there for antibodies, not for the T cell responses) shows a balanced distribution of Th1 and Th2 CD4+ T cell responses, it is not anything further to the manuscript.

We agree with the reviewer that the CD4 T cell response primarily indicates a balanced distribution and is not studied through the 3rd dose. The importance of this is that previous alum vaccines have shown a worrisome skewing to Th2 dominated response – but ours does not, likely due to the nanoparticle antigen (supported by our previous work). We have updated the text to highlight this dynamic (pg. 13-14).

“and mitigates the Th2 skewing seen in previous Alum-based vaccines”

And included the citation:

Brewer JM, Conacher M, Hunter CA, Mohrs M, Brombacher F, Alexander J. Aluminium hydroxide adjuvant initiates strong antigen-specific Th2 responses in the absence of IL-4- or IL-13-mediated signaling. *J Immunol.* 1999 Dec 15;163(12):6448-54. PMID: 10586035.

vi) One of the current challenges for the COVID-19 vaccines is immune responses in the mucosal immune response, and the authors did not show any evidence of the mucosal immune response with the current vaccine.

While we agree that mucosal immunity may be an important aspect for protection, the established correlates of protection for approval of vaccines is derived from humoral immunity. As such, we focus on the specifics of humoral immunity, and will focus on mucosal immunity in a future study.

vii) Also, the selection of macaques is poorly stated, as the vaccine targets pediatric populations, but the study was conducted using male rhesus macaques aged between 3 and 9 years. Authors need to explicitly state the age of the monkey and the age of the humans in their NHP study design and the text.

We thank the reviewer for this suggestion and have updated the text accordingly. We envision that the macaques resemble a pediatric population less so because of their age, but rather their naive status SARS-CoV-2. We have highlighted this in the text (pg 8).

“The NHPs are an antigen-naïve population, acting as a surrogate for a pediatric population“

Overall, the manuscript contains some important findings however, the lack of protection against current emerging VOCs following vaccination is a major setback for the manuscript.

We agree with the reviewer that protection is important, but we consider this outside the scope of this manuscript, particularly given the challenges with model systems (e.g. hamsters) and the currently circulating variants. The correlation between neutralizing titers and protection from infection is now well-established, including with mAb studies in humans, where only humoral immunity is provided. These points and references are included in the manuscript (pg. 15):

“In addition, clinical trials have established a correlation between anti-SARS-CoV-2 monoclonal antibody levels (i.e., humoral immunity alone) and protection from COVID-19.^{60,61} Indeed, SARS-CoV-2 variant booster vaccines have been accepted by the FDA and EMA for emergency use authorization using neutralizing antibody titer as a correlate of protection.^{62,63}”

References:

60 Schmidt, P. et al. Antibody-mediated protection against symptomatic COVID-19 can be achieved at low serum neutralizing titers. medRxiv, 2022.2010.2018.22281172 (2022).

<https://doi.org/10.1101/2022.10.18.22281172>

61 Stadler, E. et al. Monoclonal antibody levels and protection from COVID-19. medRxiv, 2022.2011.2022.22282199 (2022). <https://doi.org/10.1101/2022.11.22.22282199>

62 U.S. Department of Health and Human Services Food and Drug Administration. Emergency Use Authorization for Vaccines to Prevent COVID-19 Guidance for Industry (2022).

63 European Medicines Agency Committee for Human Medicinal Products (CHMP). Reflection paper on the regulatory requirements for vaccines intended to provide protection against variant strain(s) of SARS-CoV-2 (2022).

Accordingly, we will focus on protection studies in future work.

Reviewers' Comments:

Reviewer #1:

Remarks to the Author:

My concerns/comments have been addressed accordingly.